# The Role of the Dopamine System in Post-Stroke Mood Disorders in Newborn Rats

**DOI:** 10.3390/ijms24043229

**Published:** 2023-02-06

**Authors:** María Villa, María Martínez-Vega, Aarón del Pozo, Itziar Muneta-Arrate, Ana Gómez-Soria, Carolina Muguruza, María de Hoz-Rivera, Angela Romero, Laura Silva, Luis F. Callado, Maria José Casarejos, José Martínez-Orgado

**Affiliations:** 1Biomedical Research Foundation, Hospital Clínico San Calos-IdISSC, 28040 Madrid, Spain; 2Department of Pharmacology, University of the Basque Country, UPV/EHU, 48940 Leioa, Spain; 3Centro de Investigación Biomédic en Red de Salud Mental (CIBERSAM), 28007 Madrid, Spain; 4Neurobiology Department, IRYCIS, Hospital Universitario Ramón y Cajal, 28034 Madrid, Spain.; 5Biocruces-Bizkaia Health Research Institute, 48901 Barakaldo, Spain; 6Department of Neonatology, Hospital Clínico San Calos-IdISSC, 28040 Madrid, Spain

**Keywords:** dopamine, mood disorders, newborn, rats, stroke

## Abstract

Post-stroke mood disorders (PSMD) affect disease prognosis in adults. Adult rodent models underlie the importance of the dopamine (DA) system in PSMD pathophysiology. There are no studies on PSMD after neonatal stroke. We induced neonatal stroke in 7-day-old (P7) rats by temporal left middle cerebral artery occlusion (MCAO). Performance in the tail suspension test (TST) at P14 and the forced swimming test (FST) and open field test (OFT) at P37 were studied to assess PSMD. DA neuron density in the ventral tegmental area, brain DA concentration and DA transporter (DAT) expression as well as D2 receptor (D2R) expression and G-protein functional coupling were also studied. MCAO animals revealed depressive-like symptoms at P14 associated with decreased DA concentration and reduced DA neuron population and DAT expression. At P37, MCAO rats showed hyperactive behavior associated with increased DA concentration, normalization of DA neuron density and decreased DAT expression. MCAO did not modify D2R expression but reduced D2R functionality at P37. MCAO-induced depressive-like symptoms were reversed by the DA reuptake inhibitor GBR-12909. In conclusion, MCAO in newborn rats induced depressive-like symptoms and hyperactive behavior in the medium and long term, respectively, that were associated with alterations in the DA system.

## 1. Introduction

Arterial ischemic stroke (AIS) is considered typical of adulthood. However, the neonatal period presents a high prevalence, being a time of particular susceptibility and risk [1]. It is estimated that between two and four of every 1000 live newborns present a perinatal AIS (PAIS) [1,2]. Approximately 50% will develop some sequelae [3]. It has traditionally been considered that the sequelae of PAIS are mainly motor, although cognitive disorders or epilepsy may also appear [2]. There are very few longitudinal studies on the neuropsychiatric sequelae of PAIS and those that do exist focus on the onset of cognitive problems (memory and language) added to motor problems [4,5] or behavioral problems—especially attention deficit and hyperactivity [6]. However, post-stroke mood disorders (PSMD), in particular depression, are common in adults, demonstrable in up to 40% of cases, half of which may be major depression [7]. The development of PSMD is associated with increased morbidity and mortality in stroke patients [7,8]. Furthermore, it affects the efficacy of rehabilitation after injury due to motivational problems [8]. PSMD are not exclusively a reaction to the patient’s confirmation of the onset of a probably permanent disability but rather they are a consequence of brain damage “per se” [7,8]. Despite being so widely studied in adulthood, whether MD occurs in a similar way when stroke occurs in the perinatal period has not been investigated. It has been reported that three-to-six-year-old children with a history of severe neonatal hypoxic–ischemic encephalopathy but without cerebral palsy or other severely disabling neurosensory sequelae, present a higher incidence of depressive symptoms than a control group of healthy children without such a history [9].

Experimental models have revealed major utility in the study of PSMD in adults. AIS models using middle cerebral artery occlusion (MCAO) in adult rodents have been able to reproduce symptoms similar to that of PSMD, with depression-like symptoms appearing two to three weeks after AIS [10]. These models have succeeded in consolidating the idea that PSMD are related to an alteration in monoaminergic circuits, in relation to increased inflammation and excitotoxicity that would affect the corresponding circuits [7,8,10]. Although many studies on the involvement of monoaminergic circuits have focused on serotonin (5-HT) and norepinephrine (NE), the importance attached to alterations in dopaminergic circuits in adult depression, including post-stroke depression [11], is growing. Dopamine (DA) is the predominant catecholamine neurotransmitter in the brain; it is synthesized by neurons located in the substantia nigra and the ventral tegmental area (VTA), which project to the striatum, cortex, limbic system and hypothalamus [12,13,14]. DA binds to specific receptors, the most abundant being those called D1 and D2, whose activation increases or decreases the intracellular levels of cAMP by means of coupling to stimulatory (Gαs) or inhibitory (Gαi) G-proteins, respectively [14]. Once released into the synaptic gap, DA reuptake occurs into the presynaptic neuron via the dopamine active transporter (DAT), whose expression is altered in emotional and hyperactivity disorders [10].

The aim of this work was to assess, using a stroke model in newborn rats, whether PAIS produces PSMD in the medium and long term, focusing on the role of DA in such a condition.

## 2. Results

General data are shown in Table 1. There were no differences in weight at procedure or sex distribution. MCAO procedure led to increased mortality, approximately 15%, which was in accordance to that reported for this procedure [15].

### 2.1. MCAO-Induced Brain Injury

MCAO led to a reproducible infarct area as observed at P14 by MRI (Figure 1A). The area of infarct did not progress afterwards, so that at P37 the relative area of infarct was smaller than at P14 (Figure 1A).

MCAO-induced brain injury led to functional impairment. At P14, MCAO rats displayed impaired coordination and fine motor performance, as reflected by poorer performances on the geotaxis and grasp test, respectively (Figure 1B). In the long term, MCAO led to increased hemiparesis as shown by the CRT (Figure 1B).

MCAO led to increased cell mortality in the perilesional area, as shown by the dramatic increase in the TUNEL+ cell population (Figure 1C).

### 2.2. MCAO-Induced Mood Disorders

At P14, MCAO rats showed a longer immobilization time than SHAM animals in the TST (Figure 2A), which is interpreted as depression-like behavior [10,16].

There were no differences between groups in the swimming time in the FST at P37 (Figure 2B). In contrast, at P37 MCAO rats showed a shorter floating time than SHAM rats in the FST (Figure 2B). Floating activity reflects depression-like behavior in the FST [10,17]. In the same test, MCAO rats showed a longer climbing time (Figure 2B), which is interpreted as hyperactive behavior [10,17]. In agreement, at P37 the MCAO rats showed a greater distance covered in the OF that was associated with a larger number of entries to the center of the arena (Figure 2B), which is interpreted as hyperactive and impulsive behavior [18,19].

### 2.3. Effects of MCAO on the DA System

MCAO led to a reduction in the population of TH+ cells in the VTA as assessed at P14 (Figure 3A). This was related to reduced VTA size (median [95% CI]: 0.81 [0.67–1.07] vs. 0.53 [0.45–0.74) mm^2^ for SHAM and MCAO, respectively, U = 9, *p* < 0.05) together with reduced TH+ cell density (median [95% CI]: 339 [285–455] vs. 229 [187–310] cells/mm^2^ for SHAM and MCAO, respectively, U = 0, *p* < 0.01) in MCAO animals. At P37, although VTA size was still smaller in MCAO compared with SHAM animals (median [95% CI]: 0.79 [0.51–0.71] vs. 0.55 [0.49–0.56) mm^2^ for SHAM and MCAO, respectively, U = 8, *p* < 0.05), TH+ cell density was restored in MCAO rats (median [95% CI]: 255 [218–336] vs. 269 [241–327] cells/mm^2^ for SHAM and MCAO, respectively, U = 23, NS). As a consequence, at P37 there were no differences in the total number of TH+ cells in the VTA between SHAM and MCAO animals (Figure 3A).

DA concentration in the ipsilateral brain hemisphere was reduced in MCAO animals at P14 (Figure 3B), which was associated with a reduction in DAT expression (Figure 3C). No differences were observed in the expression of D2 receptor between SHAM and MCAO animals at P14 (Figure 3D). In contrast, at P37 brain DA concentration was higher in MCAO than in SHAM animals (Figure 3B). This was associated with reduced expression of DAT (Figure 3C) and no changes in D2 expression (Figure 3D) in MCAO compared with SHAM animals. To further account for these results, [^35^S]GTPgS binding studies were performed to assess the functionality of the D2 receptor. Such studies revealed a reduced maximal response to agonists (Emax) at the D2R of MCAO rats (median [95% CI] for ipsilateral vs. contralateral striatum: SHAM = logEC50: −6.9 [−7.5, −6.5] vs. −7.1 [−8.0, −7.0], NS, and Emax: 116.6 [114.3, 119.1] vs. 118.9 [116.7, 121.3]%, NS; MCAO = logEC50: −7.3 [−8.0, −6.5] vs. −7.3 [−7.7, −6.8], NS, and Emax: 114.4 [111.9, 117.1] vs. 125.3 [122.6,126.6]%, *p* < 0.01 by least squares regression), indicating D2 receptor hypofunction (Figure 3E).

There were no differences between SHAM and MCAO in brain concentration of serotonin (median [95% CI] for SHAM vs. MCAO: 43.2 [41.5–45.8] vs. 46.3 [41.9–53.1] nmol/g at P14, NS; 92.5 [60.5–135.4] vs. 80.5 [64.7–118.1] nmol/g at P37, NS). Brain NE concentration was increased in MCAO rats at P14 (median [95% CI] for SHAM vs. MCAO: 37.9 [35.3–40.8] vs. 49.4 [46.9–64.3] nmol/g, *p* < 0.01) but no differences between groups were observed at P37 (median [95% CI] for SHAM vs. MCAO: 45.9 [39.4–52.8] vs. 53.0 [45.8–62.0] nmol/g, NS).

### 2.4. Effects of a DA Reuptake Inhibitor

To further assess whether the depression-like behavior in MCAO rats at P14 was related to reduced brain DA concentration, some MCAO rats were treated with the DA reuptake inhibitor GBR-12909. MCAO + GBR rats did not differ from untreated MCAO rats in weight at procedure (median [95% CI]: 17.3 [16.7–17.7] g), sex distribution (Male/female: 10/9) or post-MCAO mortality (3/22; all *p* > 0.05). Treatment with GBR-12909 did not reduce the volume of damage or the MCAO-induced increase in the TUNEL+ cell population (Appendix A).

MCAO + GBR rats performed better than untreated MCAO rats in the geotaxis and grip test at P14 (Figure 4A), but at P37 the performance of MCAO + GBR on CRT was no different to that of untreated MCAO rats (Figure 4A). MCAO + GBR rats did not present depressive-like behavior at P14, showing normal performance on the TST (Figure 4B). However, at P37 the performance of MCAO + GBR rats on FST and OFT was no different to that of untreated MCAO rats (Figure 4C,D).

## 3. Discussion

In this work, the onset of depressive-like symptoms in the middle term after a stroke in newborn rats is reported for the first time. Using the TST, a behavioral paradigm to measure behavioral despair [10,16], we observed a longer immobility time in the MCAO rats, which is attributed to depressive-like symptoms in rodents. A PSMD consisting of depressive-like symptoms has been widely reported in adult rodent stroke models, appearing two to three weeks after stroke [10]. In our study, depressive-like symptoms were already evident one week after the stroke, which suggests a possible greater vulnerability in the immature brain to the development of this complication. PSMD models in adult rodents closely reproduce the pathology in humans and have been instrumental in understanding the pathophysiology of the disease and implementing therapeutic strategies [10]. The description of depressive-like symptoms is not only unprecedented in neonatal stroke animal models, but there is also no clinical evidence in this regard in human neonates. The presence of depressive symptoms has been reported in children aged three to six years old with a history of newborn hypoxic–ischemic encephalopathy [9], but this same complication has not been reported after PAIS. It is interesting that at the age at which we verified the presence of depressive-like symptoms in our work, P14, rats present a neurological development precisely comparable with that of children aged three to five years old [20].

The possible persistence of depressive-like symptoms in the long term after MCAO could not be studied using the same test, the TST. This was because excessive weight discourages performing this test in rodents [16]. Instead, we used the FST, another behavioral paradigm to measure behavioral despair [10,17]. In the FST a longer floating time corresponds to depressive-like symptoms [17]. In this case, the MCAO rats no longer exhibited depressive-like symptoms at P37. Instead, increased climbing time was observed, which is considered a trait of hyperactive behavior [17]. In agreement with this, the MCAO rats also showed greater distance covered and more entries to the center of the arena in the OFT at P37, which corresponds to hyperlocomotion and impulsive behavior, both characteristic of hyperactivity [18,19]. This is in agreement with what was previously reported in rats after neonatal acute hypoxic–ischemic insults [12,13]. The few references to mood or behavior changes after PAIS outline a higher frequency of attention deficits/hyperactivity in adolescents with a history of PAIS [6].

The depressive-like symptoms found at P14 corresponded to a reduction in the brain concentration of DA. Although monoamines traditionally associated with depression are 5-HT and NE, the importance given to the role of DA is growing, particularly in bipolar disorders [21]. In fact, symptoms such as anhedonia or amotivation are directly related to DA system dysfunction [22]. The reduction of DA concentration was related to a decreased population of DA cells in the VTA. Such a reduction can be accounted for by the exceptional vulnerability of these cells to hypoxic–ischemic damage [12,22]. The VTA is one of the main nuclei responsible for the synthesis of DA, modulating the activity of the prefrontal cortex by means of the control of striatal activity [22]. In animal models, hypodopaminergia resulting from a direct lesion of the VTA induces depressive behavior [21]. In this scenario, the reduction in DAT expression could be interpreted as an attempt to increase DA concentration by reducing its reuptake, as reported in other conditions showing decreased DA activity such as Parkinson disease [23]. The relationship between depressive-like symptoms and decreased brain DA concentration in our work was supported by the fact that the administration of a drug known for increasing brain DA concentration by inhibiting its reuptake, GBR-12909 [24], reversed said depressive symptoms. GBR-12909 inhibits dopamine uptake after blocking the dopamine carrier process at the receptor by binding to the dopamine binding site on the carrier protein itself [25]. The effects were not due to a non-specific neuroprotective effect, since GBR-12909 did not reduce the volume of the lesion or the increase in TUNEL+ cellularity in the perilesional area. No alterations in the concentration of 5-HT were detected in our study. In the case of NE, the increase observed one week after MCAO has been widely reported as an indicator of the damage that occurs after stroke [26].

The hyperactive behavior observed one month after MCAO was associated with a change in the DA system different from that observed at P14. At P37, a spontaneous recovery of DA cell density was observed in the VTA. Both an increase and a decrease in the density of DA neurons in the VTA have been reported one month after diffuse hypoxic–ischemic injury in newborn rats, depending on the severity and duration of the insult [12]. The recovery in TH+ cell density we observed was linked to an increase in DA concentration. Pharmacologic studies have provided strong evidence in adults that hyperdopaminergia leads to mania [21]. However, in models of diffuse hypoxic–ischemic brain damage in newborn rats, the long-term result is a decreased concentration of DA, although this is reported in cases where there is a decreased population density of DA cells [27]. In our work, an increased DA concentration was observed together with a reduction in DAT expression. In adult rat models of mania, hyperdopaminergia leads to increased expression of DAT as a compensatory response aimed to reduce DA neurotransmission [21]. In contrast, after neonatal diffuse hypoxic–ischemic insults, reduced expression of DAT is reported in association with decreased DA concentration [13,27]. To better understand the apparently paradoxical response in our work, we studied the functionality of D2R, which plays a major role in psychotic and non-psychotic mania associated with abnormalities in the DA system [21,28]. No differences were found between SHAM and MCAO rats in the expression levels of D2R but a decreased response to agonists was observed for D2R in the ipsilateral as compared with the contralateral striatum. Functional imaging studies have not observed differences in the density or availability of striatal D2R in non-psychotic mania in adults [21]. Studies in newborn rats exposed to diffuse cerebral hypoxic–ischemic damage report decreased expression of D2R in association with decreased DA concentration in the long term [13]. However, model studies of rotational behavior also reveal a functional imbalance of D2R between both cerebral hemispheres [13]. Our findings might suggest that the increase in DA concentration would be an attempt to compensate for the hypoactivity of D2R, with a decreased expression of DAT acting as a mechanism aimed to increase DA concentration. However, hyperactivity linked to increased dopaminergic tone is reported in KO mice with silenced DAT expression [13]. This allows us to hypothesize that the lower expression of DAT could be a primary defect after neonatal MCAO; once the population density of dopaminergic neurons normalized, this defect would lead to an increased DA concentration, which together with the functional alteration of D2R would alter the delicate balance of mesocortical circuits, leading to the hyperactive behavior. Confirmation of this speculation warrants further in-depth studies in the future. In any case, our data suggest that neonatal MCAO results in a complex imbalance in the DA system, which could be caused by the initial reduction in DA production. In an animal model, DA depletion in the neonatal period leads, two months later, to spontaneous motor hyperactivity [19].

An interesting finding is that during treatment with GBR-12909, the performance of MCAO rats in the geotaxis and grasp tests was restored. This improvement cannot be attributed to a neuroprotective effect of the treatment and was no longer observed when treatment with GBR-12909 was discontinued, since CRT performance at P37 was just as impaired in MCAO + GBR as in untreated MCAO rats. These data pave the way to the hypothesis that performance in the motor tests at P14 could have been influenced by the PSMD, that is, that the existence of a PSMD could overestimate the functional deficit after MCAO. It is known that in adults, the onset of PSMD alters the performance of patients in rehabilitation [8]. It would be very interesting to corroborate whether this also occurs after a stroke in the immature brain, since it would open up a new therapeutic possibility. There is increasing evidence of the existence of psychomotor abnormalities in diseases such as major depression are associated with an altered modulation of the dopaminergic-based subcortical–cortical motor circuit [29]. Another important aspect would be to explore the possible role of D3R in the motor improvement found after increasing the concentration of DA, since although the density of D3R in the brain is reduced and it is structurally very similar to D2R, D3R plays a specific role in the dopaminergic control of movement [14]

The main limitation of our study is that animal models, especially rodents, although widely used for this purpose [10], are not ideal for studying mood disorders. The data from this study, together with the initial data on depressive-like symptoms in children after neonatal hypoxic–ischemic damage [9], warrants further investigation on this possibility in children with a history of PAIS, although this type of study in children of that age are complicated [9].

## 4. Materials and Methods

The experimental procedures met the European and Spanish regulations (2010/63/EU and RD 53/2013) and were designed and performed by researchers qualified in laboratory animal science. The experimental protocol was approved by the San Carlos University Hospital Animal Welfare Ethics Committee (Madrid, Spain) (Protocol number: PROEX 156/19). FELASA recommendations were followed to preserve animal welfare and reduce suffering as well as the number of animals used.

### 4.1. Experimental Model

The model has been reported in detail elsewhere [15]. In short, Wistar pup rats from each litter were randomly assigned to MCAO or control. All experimental groups were gender balanced. Each MCAO animal (MCAO, n = 133) was anesthetized by sevoflurane (5% induction, 1% maintenance). The left carotid artery was dissected up to the internal and external branches division to introduce a nylon filament 0.21 mm in diameter, which was inserted 8.5–9 mm through the internal carotid artery until left MCA occlusion was attained. All procedures lasted less than 30 min during which rectal temperature was kept at 38 ± 0.5 °C using a servo-controlled heat mattress. The occlusion was maintained for 3 h. Pups were anesthetized similarly to carefully remove the filament. After sealing carotid and skin wounds the pup then returned to the dam. Control group pups (SHAM, n = 112) were similarly managed but without MCAO.

Some MCAO rats received the DA reuptake inhibitor GBR-12909 dihydrochloride (MCAO + GBR, n = 22), the well-established effect of which is to increase DA concentration in the brain [24]. A stock solution of GBR-12909 2.5 mg/mL was obtained by dissolving 50 mg of GBR 12909 in 2 mL of dimethylsulfoxide (Sigma Aldrich, St. Louis, MO, USA), then adding 18 mL of 20% Captisol^®^ (Abmole Bioscience, Dallas, TX, USA) and storing at −20 °C. An appropriate volume of the stock solution was further diluted in 0.9% saline to administer 5 mg/kg in 100 µL to each rat in a daily i.p. injection from P11 to P17.

The number of animals for each experiment is shown in Appendix A.

### 4.2. Functional Studies

A set of sensorimotor and cognitive tests was performed at P14 or P37 [15,30,31]. All tests were video recorded to be assessed by three different examiners blinded to the experimental group. In short, the tests performed at P14 were the inverse geotaxis (coordination: time required to turn 180° after being placed downwards on a ramp tilted at 45°) and grip tests (strength: grasp reflex score after leaning a thin rod against each paw palm); the test at P37 was the cylinder rearing test ([CRT] hemiparesis: after placing the rat in a methacrylate transparent cylinder, 20 cm diameter and 30 cm height, initial forepaw preference [left, right or both] was counted during a 3 min trial [minimum of four wall contacts]).

### 4.3. Studies on Mood Disorders

Another set of tests was performed to assess PSMD. All tests were video recorded and assessed by two researchers unaware of the experimental group.

At P14:

Tail Suspension Test (TST): Evaluates depressive symptoms in response to a despair situation [16]. Immobility time was measured over a period of 6 min in which the rat was strained by the tail, which prevented it from grasping any object.

At P37:

Forced Swimming Test (FST): Evaluates depressive or hyperactive behaviors in rats exposed to a despair situation [17]. Each rat was placed in a methacrylate cylinder of 20 cm in diameter and 50 cm in height, filled with water at a controlled temperature of 24 °C (±1 °C), up to a height of 30 cm. After an initial day of habituation, keeping the animal in the cylinder for 15 min, the next day they were kept for 5 min. During this period, the predominant behavior was evaluated every 5 s, differentiating between floating (total absence of movement), swimming (horizontal movement) and climbing (vertical movement on the cylinder wall), calculating the ratio of each to the total. The water was changed between each animal, both in the familiarization phase and in the test phase.

Open Field Test (OFT): Evaluates hyperlocomotion and impulsivity [18]. The rats were placed in an opaque methacrylate box, measuring 40 cm × 40 cm and with a black floor, for 10 min. The entries to the center of the field were counted and the distance traveled by each animal was analyzed using EthoWatcher^®^ software (UFSC, Santa Catarina, Brazil).

### 4.4. Measurement of the Extent of Brain Injury by MRI

MRI was performed at P14 and P37 at the BioImaC (Universidad Complutense, Madrid, Spain), a node of the ICTS ReDiB, using a 1 Tesla benchtop MRI scanner (Icon (1T-MRI); Bruker BioSpin GmbH, Ettlingen, Germany) to obtain T2WI slices. The technical specifications as well as the protocol to measure the volume of brain tissue loss and perilesional increased intensity area are reported elsewhere [15,30]. Areas of interest were manually delineated using ImageJ 1.34 s software (NIH, Bethesda, MD, USA).

### 4.5. Sampling

After the MRI studies, rats were sacrificed using a lethal injection of diazepam and ketamine. Some rats were transcardially perfused with cold saline and then cold formalin (4%), their brains then harvested and stored in 4% formalin to perform histologic studies. Other rats were transcardially perfused with cold saline, their brains then harvested, snap frozen in isopentane and stored at −80 °C to perform biochemical studies.

For [^35^S]GTPγS binding assays, fresh brain samples were obtained at P37 without prior perfusion.

### 4.6. Histologic Studies

Histologic studies were performed in paraffin-embedded coronal sections (4 µm) obtained at a level corresponding to plates 31 and 38 for the TUNEL assay and for TH staining of the Paxinos and Watson Atlas, respectively [32].

At P14, brain sections were stained with TUNEL (ApopTag^®^ In Situ Apoptosis Detection Kit, Promega, WI, USA) to identify cell death as reported elsewhere [15,30,31]. Microphotographs from three areas (250 × 250 µm) at the ipsilateral parieto–temporal cortex adjacent to the infarct area were obtained using a Leica TCS SP5 confocal microscope system (Leica, Wetzlar, Germany). Histologic studies were performed by a researcher blinded to the experimental group.

Immunohistochemical studies to detect the presence of DA neurons in the VTA at P14 and P37 were performed as reported elsewhere [15,30,31]. DA neurons were labeled with a primary anti-TH antibody (1:500, SantaCruz Biotechnology, Dallas, TX, USA), kept at 4 °C overnight and then ImmPRESS Universal Antibody Polymer Reagent was added (Vector Laboratories, Burlingame, CA, USA) for 30 min to label the primary antibody with horseradish peroxidase (HRP). Subsequently, the presence of DA cells was revealed by the use of diaminobenzidine (DAB) (Vector Laboratories, Burlingame, CA, USA). Counterstaining with Nissl was performed by immersion in 0.5% toluidine blue (Sigma Aldrich, St. Louis, MO, USA) to locate cell nuclei. Microphotographs of the left VTA were obtained at a level corresponding to plate 38 of the Paxinos and Watson Atlas [32] using an optical microscope (Leica Biosystems, Nussloch, Germany) at 10× magnification. VTA size was quantified on the photomicrograph by manually outlining the area using ImageJ software. The number of anti-TH-antibody-positive cells present within the boundaries of the VTA was counted.

### 4.7. Western Blot Studies

Western blot analysis was performed on ipsilateral brain striatum samples containing 20 µg of total protein at P14 and P37 as reported elsewhere [31]. To quantify the expression of the DA active transporter (DAT) and that of D2 receptors (D2R), anti-DAT or anti-D2R primary antibodies already labeled with HRP (1:200, SantaCruz Biotechnology, Dallas, TX, USA) were used; membranes were developed using an enhanced chemiluminescence (ECL) reagent (Abbkine, Wuhan, China) in a Syngene GBOX ChemiXX6 developer (Fisher Scientific, Newington, NH, USA). Protein levels were expressed as ratio of protein measurement/β-actin.

### 4.8. Monoamine Content Measurement

DA, 5-HT and NE concentrations were measured in brain samples obtained at P14 and P37, using HPLC coupled to an ESA coulochem detector in the neurobiology laboratory of the Ramón y Cajal Institute for Health Research, located in the Neurobiological Research Department of Ramón y Cajal Hospital. (Madrid, Spain). During the entire process, samples were kept cold (ice at 4 °C) to avoid oxidative processes; 0.4 N perchloric acid was added to the tissue sample in a ratio of 6 µL for each mg of tissue, which was homogenized by sonication and then centrifuged at 11,000× *g* at 4 °C for 20 min. The supernatant was passed through 0.45 µm filters and stored at −80 °C until HPLC was performed. Conditions were: one column (Nucleosil 5C18), the mobile phase, a 0.1 M citrate/acetate buffer, pH 3.9 with 10% methanol, 1 mM EDTA and 1.2 mM heptane sulfonic acid; the voltage conditions of the detector were: D1 (+0.05), D2 (−0.39) and the guard cell (+0.40). Levels relative to the hemisphere ipsilateral to the lesion were analyzed. The results obtained were expressed in nmol/g of tissue.

### 4.9. [^35^S]GTPγS Binding Assays

The process has been reported in detail elsewhere [28]. In short, for brain cellular membrane preparation the left and right striatum were dissected and immediately stored at −70 °C until assay. For each hemisphere, tissue samples from same experimental groups (SHAM and MCAO) were pooled to obtain the enriched fractions of plasma membranes. Tissue samples were homogenized with a Teflon-glass grinder (IKA Labortechnik, Satufen, Germany) in 30 volumes of homogenization buffer (1 mM EGTA, 3 mM MgCl_2_, 1 mM DTT, and 50 mM Tris-HCl, pH 7.4) supplemented with 0.25 M sucrose. Samples of the membrane-enriched fraction (P2 fraction) containing 0.5 mg protein were centrifuged in a benchtop centrifuge (EBA 12 R, Hettich Instruments, Tuttlingen, Germany) at 14,000 rpm for 15 min at 4 °C and stored at −80 °C. On the day of the experiment the membrane pellets were defrosted (4 °C), thawed and re-suspended in incubation buffer containing 1 mM EGTA, 3 mM MgCl_2_, 100 mM NaCl and 50 mM Tris-HCl, pH 7.4. A final protein concentration of approximately 0.08 mg/mL was attained.

[^35^S]GTPγS binding assays were performed in 96-well plates in a final volume of 250 μL, containing 1 mM EGTA, 3 mM MgCl_2_, 100 mM NaCl, 0.2 mM DTT, 50 μM GDP, 50 mM Tris–HCl at pH 7.4 and 0.5 nM [^35^S]GTPγS. Increasing concentrations of the D2R agonist *N*-propylapomorphine (NPA,10-4M-10-10M; seven concentrations by quadruplicate; three independent experiments) were incubated to perform the stimulation curves. In addition, the D2R antagonist haloperidol (10 µM) was co-incubated with a saturating concentration of the agonist NPA (10 µM) to test the specificity of D2-receptor-dependent G-protein activation. Incubations were commenced by adding the membrane suspension (20 μg of membrane proteins per well) at 30 °C for 120 min with shaking (450 rpm). Incubations were discontinued by rapid filtration under vacuum (1450 FilterMate Harvester, PerkinElmer, Waltham, MA, USA) through GF/C glass fiber filters (Printed Filtermat A) pre-soaked in ice-cold incubation buffer. Filters were then rinsed three times with 300 μL ice-cold incubation buffer, air dried (20 °C, 120 min) and counted for radioactivity (4 min) by liquid scintillation spectrometry (MicroBeta TriLux counter, PerkinElmer). Non-specific binding of the radioligand was defined as the remaining [^35^S]GTPγS binding in the presence of 10 μM unlabeled GTPγS and the basal binding as the signal in the absence of agonist.

### 4.10. Statistical Analysis

Data were analyzed using GraphPad Prism™ software version 9.1 (GraphPad Software, San Diego, CA, USA). The pharmacological parameters of the stimulation curves of the [^35^S]GTPγS binding, Emax (maximal effect) and EC50 (drug concentration that determines the half maximal effect) were expressed as means ± SEM, which were calculated by non-linear analysis and compared by a co-analysis of the curves (F-test). For the remaining data, after assessing the normality of data distribution using the D’Agostino–Pearson test, data revealed a non-normal distribution. Therefore, data were shown as median (95% CI) and compared using the Mann–Whitney test or Kruskall–Wallis with Dunn’s test for multiple comparisons. A value of *p* < 0.05 was considered statistically significant.

## 5. Conclusions

In conclusion, MCAO-induced stroke in newborn rats resulted in PSMD consisting of depressive-like symptoms in the medium term and then hyperactive behavior in the long term. Those PSMD were related, in the first case, to a decreased cerebral concentration of DA due to the decreased density of DA neurons in the VTA. Thus, treatment with a drug that increases the brain concentration of DA reversed these symptoms and even improved motor performance. In the long term, hyperactive behavior was associated with hypofunction of striatal D2R. These data suggest the need to investigate the onset of PSMD after stroke in newborns and present the DA system as a possible new therapeutic target for the holistic treatment of such a disease.

## Figures and Tables

**Figure 1 ijms-24-03229-f001:**
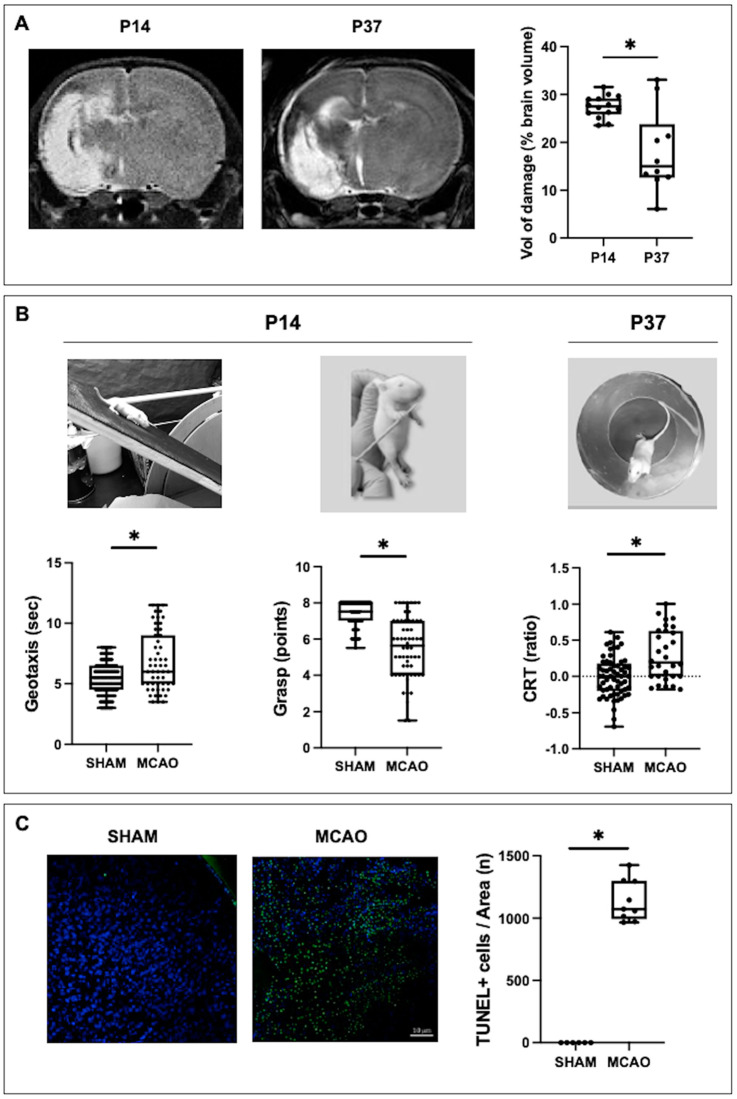
Assessment of brain damage in rats submitted to MCAO at seven days of life (P7) and the corresponding healthy controls (SHAM). (**A**) Representative T2-Weighted MRI scans, and quantification of lesion volume, measured at P14 and P37. (**B**) Functional consequences of MCAO measured in the medium (P14) and long terms (P37). (**C**) Illustrative photomicrographs of TUNEL staining in brain samples obtained at P14 and their quantification. Original magnification: ×20. Boxes represent the median and the interquartile range, whereas whiskers represent the minimum and maximum values in each group. CRT: cylinder rear test. (*) *p* < 0.05 by Mann−Whitney test: (**A**): U = 20, *p* = 0.18; (**B**): *Geotaxis:* U = 2516, *p* = 0.0006; *Grasp:* U = 1412, *p* < 0.0001; *CRT:* U = 480, *p* = 0.0001; (**C**): U = 0, *p* = 0.0002.

**Figure 2 ijms-24-03229-f002:**
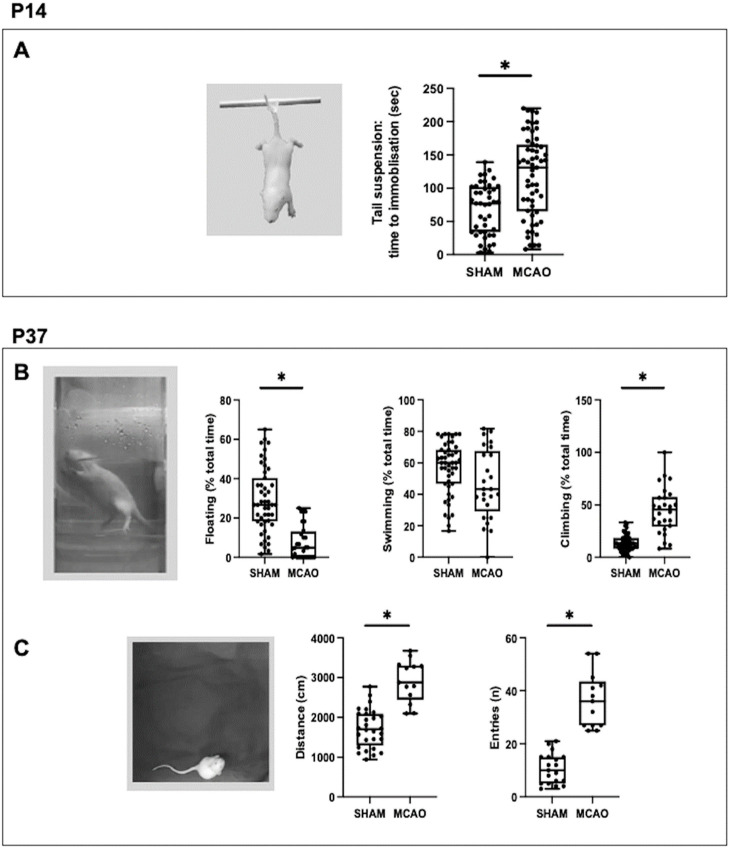
Assessment of mood disorders in rats submitted to MCAO at seven days of life (P7) and the corresponding healthy controls (SHAM). (**A**) Quantification of immobilization time in the tails suspension test, performed at P14. At P37, quantification of (**B**) floating, swimming and climbing time in the forced swimming test, and (**C**) distance traveled and number of entries to the center of the arena in the open field test. Boxes represent the median and interquartile range, whereas whiskers represent the minimum and maximum values in each group. (*) *p* < 0.05 by Mann–Whitney test: (**A**): U = 624, *p* < 0.0001; (**B**): *Floating:* U = 149, *p* < 0.0001; *Swimming:* U = 210, *p* = 0.096; *Climbing:* U = 103, *p* < 0.0001; (**C**): *Distance*: U = 17, *p* < 0.0001; *Entries*: U = 0, *p* < 0.0001.

**Figure 3 ijms-24-03229-f003:**
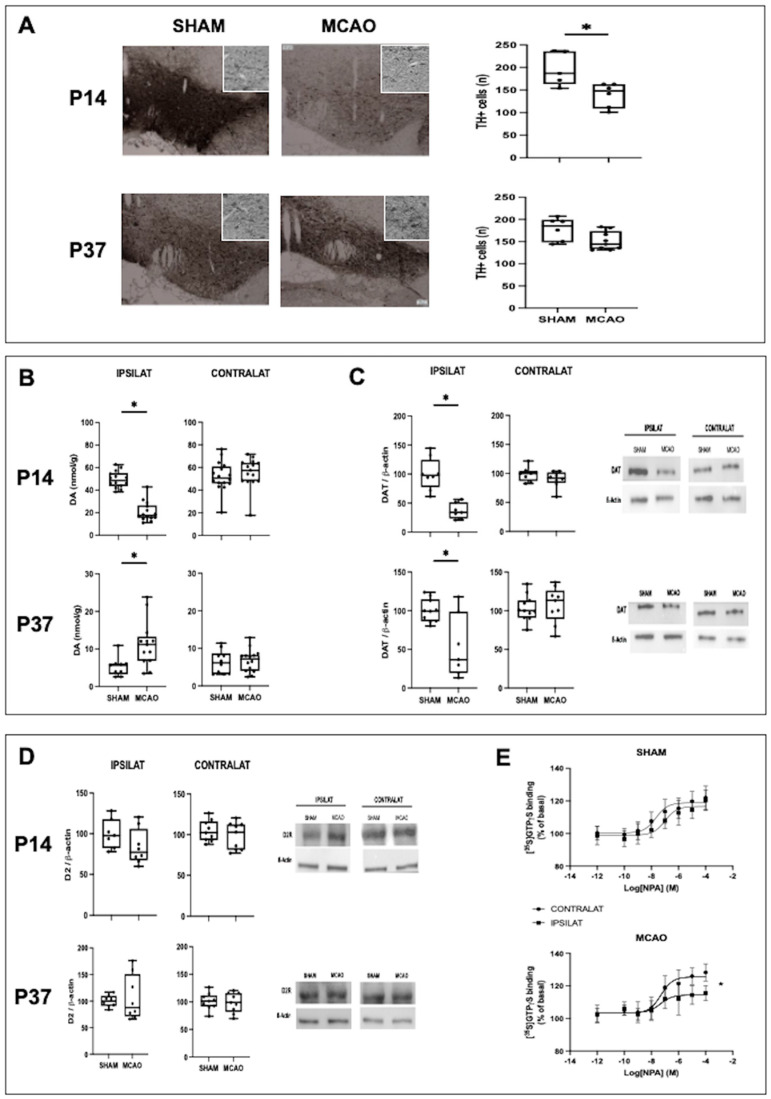
Assessment of the dopamine system in rats submitted to MCAO at seven days of life (P7) and the corresponding healthy controls (SHAM). (**A**) Illustrative photomicrographs of the ipsilateral ventral tegmental area (VTA) after thyroxin hydroxylase (TH) staining and the quantification of the number of TH+ cells in VTA. Original magnification: low magnitude = ×10; high magnitude = ×20. (**B**) Quantification of dopamine (DA) concentration (high-performance liquid chromatography). (**C**) Quantification of dopamine transporter (DAT) expression with representative examples of Western blot studies. (**D**) Representative samples of Western blot studies on D2 receptor expression performed in samples from the ipsilateral striatum and the corresponding graphical representation of the densitometric analysis. (**B**–**D**) Studies were performed on samples from the ipsilateral striatum. Boxes represent the median and the interquartile range, whereas whiskers represent the minimum and maximum values in each group. (*) *p* < 0.05 by Mann−Whitney test: (**A**): P14: U = 0, *p* = 0.004; *P37:* U = 24, *p* = 0.77; (**B**): *P14:* U = 3, *p* < 0.0001; *P37:* U = 18, *p* = 0.005; (**C**): *P14:* U = 0, *p* = 0.0003; *P37:* U = 12, *p* = 0.02; (**D**): *P14:* U = 14, *p* = 0.12; *P37:* U = 27, *p* = 0.64. (**E**) Concentration-response curves of the D2 receptor agonist NPA stimulated [^35^S]GTPγS specific binding over basal in ipsilateral (IPSILAT) and contralateral (CONTRALAT) striatum; points represent mean (standard error of mean). (*) *p* < 005 by least squares regression.

**Figure 4 ijms-24-03229-f004:**
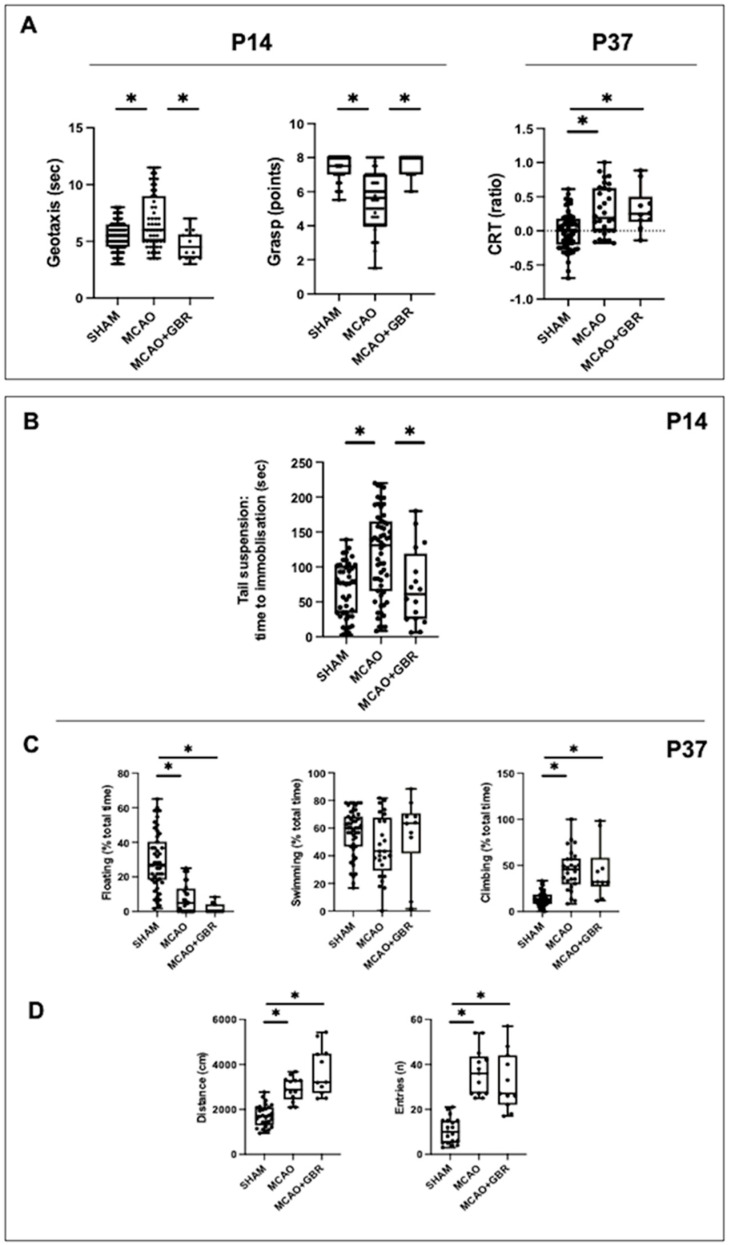
Assessment of functional deficits and mood disorders in rats submitted to MCAO at seven days of life (P7) and then treated with vehicle (MCAO) or the DA reuptake inhibitor GBR12909 (MCAO + GBR) and the corresponding healthy controls (SHAM). (**A**) Functional consequences of MCAO measured in the medium (P14) and long terms (P37). (**B**) Quantification of immobilization time in the tails suspension test, performed at P14. At P37, quantification of (**C**) floating, swimming and climbing time in the forced swimming test, and (**D**) distance traveled and number of entries to the center of the arena in the open field test. Boxes represent the median and the interquartile range, whereas whiskers represent the minimum and maximum values in each group. CRT: cylinder rear test. (*) *p* < 0.05 by Kruskal−Wallis test with Dunn’s test for multiple comparisons: (**A**): *Geotaxis:* W = 15, *p* = 0.0005; *Grasp:* W = 61, *p* < 0.0001; *CRT:* W = 170, *p* = 0.0001; (**C**): U = 0, *p* = 0.0002. (**B**): W = 19, *p* < 0.0001; (**C**): *Floating:* W = 44, *p* < 0.0001; *Swimming:* W = 3, *p* = 0.18; *Climbing:* W = 39, *p* < 0.0001; (**D**): *Distance*: W = 33, *p* < 0.0001; *Entries*: W = 30, *p* < 0.0001.

**Table 1 ijms-24-03229-t001:** General data.

Group	Weight at Procedure (g) ^1^	Male/Female	Post-MCAO Mortality
SHAM	17.4 [17.1–18.0]	51/59	2/112
MCAO	17.7 [17.1–18.3]	56/56	21/133 *

MCAO: middle cerebral artery occlusion. ^1^ Median [95% CI] *: X^2^ = 14.0, *p* < 0.05.

## Data Availability

The datasets generated during and/or analyzed during this study are available from the corresponding author on reasonable request.

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
