# Peer review of "The Role of the Dopamine System in Post-Stroke Mood Disorders in Newborn Rats"

_ijms, 2023, doi:10.3390/ijms24043229_

Round 1
Reviewer 1 Report
Overall the data presented represents valuable information regarding the role of dopamine in regulating post-stroke mood disorders. This is a very important issue because of the frequency of perinatal arterial ischemic stroke in live newborns. The authors have planned well and presented the data in a way that will help the scientific community focus on the dopaminergic system.
The authors have stated that treatment with a drug that increases the brain concentration of DA reversed the symptoms and improved motor performance. Did the authors study the expression of DR3 as motor function is associated with D3 receptor`?
Author Response
The authors have stated that treatment with a drug that increases the brain concentration of DA reversed the symptoms and improved motor performance. Did the authors study the expression of DR3 as motor function is associated with D3 receptor?
This is a very interesting question, and we thank the Reviewer for the comment, which opens exciting perspectives for the future. In this first study we have been focused on D2 receptors, but we acknowledge the possible role of D3R. We have included a sentence in this regard in the new Discussion (lines 327-331)-
Regarding English language, the manuscript has been carefully reviewed by a professional, native English editor.
Reviewer 2 Report
The author used a model of stroke in newborn rats to evaluate the depressive-like symptoms in the medium and long term and its association with alterations of the DA system.
1) Figure 3A, can the author show better images of both low magnitude and high magnitude?
2) Figure 3B-D, the results only show the ipsilateral. Can the author also show the results of contralateral?
Author Response
We wish to thank the Reviewer for their comments, which have enriched the quality of the manuscript.
1) Figure 3A, can the author show better images of both low magnitude and high magnitude?
Images of high amplitude have been included in Figure 3A. Image quality has been improved in all Figures.
2) Figure 3B-D, the results only show the ipsilateral. Can the author also show the results of contralateral?
Results from contralateral side have been included in Fugures 3B-D.
The English language of the manuscript has been carefully reviewed by a professional, native English editor.
Reviewer 3 Report
The study performed by authors “ Role of dopamine system in post-stroke mood disorders in 2 newborn rats”, there are some points to be further addressed. Most of them are related to research topics and experimental results.
1-First of all, the language and scientific writing needs significant improvement to ensure clarity of the message and appropriate reporting of scientific detail.
2-The image resolution is low. Prepare tables and figures according to the journal guidelines. They are not clear. Please improve the quality of the images
3-The research objectives of this study are not clearly addressed throughout this manuscript, and the results and conclusions need to be better presented.
4- What was the rationale behind the selection of DA reuptake inhibitor (GBR-12909 ) in the present study?
5- To better identify the key amino acids involved in the active site of the protein, it is suggested to use molecular docking between the inhibitor GBR-12909 and the target.
Author Response
We wish to thank the Reviewer for their comments, which have enriched the quality of the manuscript.
1-First of all, the language and scientific writing needs significant improvement to ensure clarity of the message and appropriate reporting of scientific detail.
We agree with the Reviewer. The English language of the manuscript has been carefully reviewed by a professional, native English editor. In addition, instead of a "Results and Discussion" section, in the revised version of the manuscript we offer an independent, more academic, explanatory and referenced Discussion section. In this way, we believe we have improved the clarity of the message and the quality of the interpretation of the results.
2-The image resolution is low. Prepare tables and figures according to the journal guidelines. They are not clear. Please improve the quality of the images
The quality of all images and figures has been improved to meet the standards of the journal (600 dpi for eacf figure).
3-The research objectives of this study are not clearly addressed throughout this manuscript, and the results and conclusions need to be better presented.
As we have said, we present in this revised version of the manuscript a Discussion in which the objectives of the study are clarified and the results and their implications are discussed in depth.
4- What was the rationale behind the selection of DA reuptake inhibitor (GBR-12909 ) in the present study?
We used GBR-12909, a drug that increases brain dopamine concentration, to test the hypothesis that depressive-like symptoms were related to a decrease in dopamine concentration. This aspect is better explained in the new Discussion of the revised version of the manuscript (lines 266-269).
5- To better identify the key amino acids involved in the active site of the protein, it is suggested to use molecular docking between the inhibitor GBR-12909 and the target.
Although this aspect is certainly interesting, it is beyond the limits of our study. In addition, the mechanisms of action of GBR-12909 have already been studied in depth. We add a reference to this effect in the new Discussion of the revised version of the manuscript (lines 269-271, ref. 25)